# Do We Actually Help Choking Children? The Quality of Evidence on the Effectiveness and Safety of First Aid Rescue Manoeuvres: A Narrative Review

**DOI:** 10.3390/medicina60111827

**Published:** 2024-11-07

**Authors:** Jakub R. Bieliński, Riley Huntley, Cody L. Dunne, Dariusz Timler, Klaudiusz Nadolny, Filip Jaskiewicz

**Affiliations:** 1Department of Emergency Medicine and Disaster Medicine, Medical University of Lodz, 90-419 Lodz, Poland; jakub.bielinski@umed.lodz.pl (J.R.B.); dariusz.timler@umed.lodz.pl (D.T.); 2School of Nursing, Faculty of Applied Science, University of British Columbia, Vancouver, BC V6T 2B5, Canada; rileydj@student.ubc.ca; 3Department of Emergency Medicine, University of Calgary, Calgary, AB T2N 1N4, Canada; cody.dunne@ucalgary.ca; 4Department of Community Health Sciences, University of Calgary, Calgary, AB T2N 1N4, Canada; 5Department of Emergency Medical Service, Faculty of Medicine, Silesian Academy in Katowice, 40-555 Katowice, Poland; prrm.knadolny@interia.pl

**Keywords:** first aid, paediatric, children, foreign body airway obstruction, choking, airway, back blows, chest thrusts, abdominal thrusts, Heimlich manoeuvre

## Abstract

The management of foreign body airway obstruction has evolved over time from back blows and chest thrusts to abdominal thrusts. However, current guidelines worldwide are based on outdated data, with unclear evidence regarding the effectiveness and safety of these rescue manoeuvres. Concerns persist about the potential of these techniques to cause injury, especially in children; therefore, a critical revision to ensure optimal child safety is necessary. The literature on first aid for paediatric choking was identified through the searching of various databases. Studies were evaluated for their relevance, quality, and currency. The analysis examined guideline consistency with current evidenced-based medicine and identified research gaps. The analysis of the available data was supplemented by adult-based evidence due to the scarcity of paediatric-specific research. First aid guidelines and recommendations for paediatric choking are divergent and generally grounded in low-quality evidence derived primarily from case studies. Studies since 2015 have shown highly diverse methodologies and often lack details on the execution of individual techniques, body positioning or the specific characteristics of study groups, which are crucial when comparing the effectiveness and safety of rescue manoeuvres. Updating evidence-based scientific knowledge for future recommendations is crucial.

## 1. Introduction

### 1.1. Background

Children are particularly vulnerable to foreign body airway obstruction (FBAO) due to their unique anatomical features, including smaller airways and disproportionately larger tongues [1]. This not only increases the risk of blockage by small objects but also results in lower cough-generating forces, as explained by the Hagen–Poiseuille law, which further elevates the risk of complete airflow obstruction [2]. Additionally, during early development, children continuously explore their environment and learn about the world through senses such as touch and taste [2,3]. This exploratory behaviour often involves placing objects in their mouths, further increasing the risk of airway blockage. These factors contribute to FBAO being one of the leading causes of accidental deaths in children under the age of 16.

Bystander intervention in the form of first aid rescue manoeuvres is the primary method to treat severe FBAO, and clearing the obstruction before the arrival of emergency medical personnel is the strongest predictor of survival [1,2,3,4,5]. However, recent data suggest that up to half of these deaths occur with no action taken, indicating bystanders lack either proficiency or confidence using these manoeuvres, resulting in significant delays to proper care [1,2,3,4,5,6,7]. Untreated FBAO-induced hypoxia rarely occurs without serious neurological damage [8]. Individuals who experience prolonged airway obstruction are exposed to an increased risk of severe neurological impairment or brain death [4,5]. The choking person is also at risk of other life-threatening complications such as swelling and inflammation of the airway which occurs in 5% of cases after the delayed removal of a foreign body, even from a partial obstruction [1]. As complete blockage results in respiratory failure and cardiac arrest, the survival rate for paediatric out-of-hospital cardiac arrest due to FBAO remains very low [8,9].

### 1.2. The Past

Since the early days of first aid, managing a choking victim with a FBAO has conventionally involved the administration of back blows, also known as back slaps. Reports dating back to the 17th century describe attempts to aid choking individuals with “concussions to the body”, which likely represent an early use of this rescue manoeuvre. Supposedly, back blows were used from time immemorial in conjunction with manual efforts to dislodge the object, inversion or succussion of the body, and percussion of the chest, a practice that also bears resemblance to modern chest thrusts [10,11]. While other manoeuvres faded over time due to ineffectiveness or safety concerns, back blows and chest thrusts persisted and were formally recommended as safe and effective by some of the earliest providers of resuscitation guidelines [10,11,12].

Uncertainty has arisen over time, however, and medical professionals have commenced debating on the relative benefits and risks of administering back blows and chest thrusts to a choking individual [10,12,13,14,15]. One vocal dissident to back blows and chest thrusts was Dr Henry J. Heimlich. Between 1974 and 1979, he cautioned against using back blows, which he referred to as “death blows”, stating in his works that they could potentially result in a foreign object becoming lodged deeper into the airway, causing a hopeless blockage [16,17]. In 1987, he famously claimed that chest thrusts were “hazardous, even lethal” due to their functional similarity to chest compressions performed during cardiopulmonary resuscitation (CPR) and might lead to the same trauma, especially in children [11]. Dr Heimlich introduced abdominal thrusts, which were later named after him, as an alternative technique. By 1986, both the American Heart Association (AHA) and the American division of the International Federation of Red Cross and Red Crescent Societies (IFRC) had incorporated the abdominal thrusts into their guidelines, ceasing to recommend back blows and chest thrusts, which were either considered a “last, desperate effort” or discouraged altogether [10]. This shift was influenced by Dr Heimlich’s promotion of abdominal thrusts, based on unsupported cases from lay press reports, as he sought to discredit AHA experts who still endorsed alternative techniques [10,18]. The Heimlich manoeuvre became a widely recognised and commonly recommended response for FBAO in both children and adults.

### 1.3. The Present

Nowadays, guidelines around the world indicate a lack of clarity among experts regarding the first aid management of paediatric FBAO. The International Liaison Committee on Resuscitation (ILCOR), a committee dedicated to promoting the implementation of evidence-informed first aid, continues to synthesise contemporary data on FBAO management. Despite this, committees such as the European Resuscitation Council (ERC), the American Heart Association (AHA), the Australian and New Zealand Committee on Resuscitation (ANZCOR), the Resuscitation Council of Asia (RCA), St John Ambulance (SJA), and similar organisations have discordant guidelines regarding recommended manoeuvres. Abdominal thrusts continue to be a commonly advocated technique, although a significant number of organisations either completely reject them or include them in an algorithm that also consists of back blows and chest thrusts, which are once again gaining popularity [3,19,20,21,22,23]. These variations highlight the complexities in establishing uniformly accepted recommendations based on weak data. The inconsistencies are especially noticeable in the recommendations for treating children older than 1 year old who are conscious with an ineffective cough, as shown in Table 1.

Despite efforts to standardise guidelines, some countries have multiple recommendations from national organisations. Canada is a unique case where five nationally recognised first aid and CPR training agencies have developed three distinctly different FBAO approaches, as shown in Table 2 [24,25,26,27,28]. To standardise first aid training, the Canadian Guidelines Consensus Task Force was formed in 2015, leading to the first publication of the Canadian Consensus Guidelines on First Aid and CPR in 2016 [29]. However, a literature search has shown no subsequent publications of these guidelines. As international organisations like the IFRC and ILCOR release new science and best practice recommendations, Canadian training agencies have returned to individually updating their training programmes, resulting in practice inconsistencies once again [28]. This scenario is not unique to Canada and reflects challenges shared by regions and countries without a dedicated, single resuscitation council [30].

Even in the presence of resuscitation councils, inconsistencies in recommendations versus practice can occur when additional authoritative bodies (e.g., government agencies managing workplace health and safety) develop their own standards which much be followed by companies and workers to adhere to local regulations. Although first aid and CPR programmes may have regional or nationally approved training curricula, these authoritative bodies may mandate standardised approaches within their jurisdictions. For instance, in the province of Ontario, Canada, five back blows are alternated with five abdominal thrusts to receive programme accreditation [26]. This jurisdictional variation adds another layer of complexity and can lead to further discrepancies in training and practice within the same first aid and CPR training organisation. In the absence of resuscitation councils, health service organisations, medical colleges or licencing boards, and emergency medical services (e.g., ambulance services) must develop their own guidelines, recommendations, standards, or protocols to address these inconsistencies. This decentralised approach increases the risk of disparities in methods and practices. These challenges underscore the need for high-quality studies to inform recommendations for emergency procedure based on robust scientific evidence. Presently, the limited available evidence restricts ILCOR’s ability to make strong treatment recommendations, resulting in significant disparity across various nations and councils as they must translate weak evidence into practice. Ongoing efforts to harmonise first aid and resuscitation practices globally must focus on improving the quality of the evidence base [31,32].

### 1.4. The Future

The effectiveness of life-saving manoeuvres in the first aid management of paediatric FBAO has limited contemporary evidence. According to the ERC, “age-specific manoeuvres for FBAO have been part of resuscitation guidelines for more than 25 years” [3]. Yet, according to the ILCOR update on the issue, their recommendations are “weak” and based on evidence with a quality of “very low certainty” [33,34]. It is important to note that much of ILCOR’s research is from several decades ago, with most of the referenced material originating from the previous century, which raises concerns about its current validity and relevance [3,33,34]. Therefore, it is accurate to state that there is a lack of knowledge about the effectiveness and safety of the recommended rescue manoeuvres, the balance of which is crucial when deciding which to recommend.

The issue of paediatric FBAO and the rescue manoeuvres for choking victims is alarming due to the high incidence of choking incidents, significant mortality risk and limited first aid provided in such cases [1,2,33]. The increased vulnerability of children to severe upper airway obstruction is compounded by the challenges in performing effective rescue manoeuvres. The disproportion between the size of an adult hand and a paediatric torso, as demonstrated later in the figures, highlights the risk of injury to a child’s fragile body when forceful procedures are used. Yet, uncertainties remain regarding the evidence and reliance on outdated references, resulting in a lack of universal agreement on first aid recommendations and guidelines.

Consequently, to ensure the safety and well-being of children, an in-depth analysis is required to critically re-evaluate the life-saving manoeuvres recommended by the various global resuscitation guideline developers in the first aid management of paediatric FBAO. This analysis emphasises its most disputed and controversial section: treating children who are conscious and have an ineffective cough (otherwise known as a complete or severe obstruction). Therefore, we aim to conduct a narrative review of the recent literature, assessing the effectiveness and safety of standard first aid rescue manoeuvres for children aged one year or older—back blows, chest thrusts, and abdominal thrusts—and identify the gaps in evidence and knowledge in order to outline the direction of future research.

## 2. Materials and Methods

### 2.1. Identification of Relevant Guidelines and Literature

After we carried out a search of the websites of resuscitation councils and organisations focused on first aid and resuscitation guidelines, in order to identify further literature, including those from ERC, AHA, IFRC, MFMER, ANZCOR, JRC, KACPR, RCA, RCSA, SJA, RCUK, CRC, CSP, HSF, RLSS, and SJA Canada, we identified studies describing the effectiveness and safety of paediatric FBAO manoeuvres. To conduct this review, we searched databases including PubMed (National Center for Biotechnology Information, National Library of Medicine, 8600 Rockville Pike, Bethesda, MD 20894, USA), Google Scholar (Google LLC, 1600 Amphitheatre Parkway, Mountain View, CA 94043, USA) and CORE (The Open University, Walton Hall, Milton Keynes, MK7 6AA, United Kingdom) up until July 2024. We used a combination of keywords such as back blows, chest thrusts, abdominal thrusts, foreign body airway obstruction, choking, and foreign body aspiration. Articles were included based on specific criteria, namely relevance to first aid procedures for paediatric FBAO, methodological quality of the research, and recency of publication, with a focus on articles published from 2015. Finally, we included studies of adults for a more comprehensive analysis of the data, due to the limited quantity and quality of paediatric evidence. This process is represented in Figure 1 by a PRISMA (Preferred Reporting Items for Systematic Reviews and Meta-Analyses) flow diagram.

### 2.2. Analysis and Synthesis

One researcher reviewed all identified articles and guidelines to assess whether they met the inclusion criteria. After screening the abstracts, we conducted a full-text review to determine eligibility based on predefined factors such as relevance to the topic, study design, and quality of evidence. We extracted relevant data, including the effectiveness, safety, and application of various rescue manoeuvres, and compiled them into a narrative summary for each technique. We then compared the scientific evidence with guidelines and recommendations from different regions, highlighting the degree of alignment between them. All researchers reviewed the findings together, discussing and identifying research gaps. We synthesised the literature to provide a comprehensive summary of rescue manoeuvres, noting any deficiencies, contradictions, or areas requiring further investigation.

## 3. Results and Discussion

### 3.1. Back Blows

Back blows are forceful slaps with the heel of the hand between the choking person’s shoulder blades while leaning them over at the waist to face the ground, as presented in Figure 2. Back blows are widely believed to create a strong air vibration and increase the intrathoracic pressure in the airway, which helps to dislodge the obstruction [35]. Although this mechanism seems reasonable, it has not been clearly proven by any identified study.

Back blows are recommended by the ERC, the ANZCOR, the IFRC, the Mayo Foundation for Medical Education and Research (MFMER), the Resuscitation Council UK (RCUK), the Japan Resuscitation Council (JRC) and the Korean Association of Cardiopulmonary Resuscitation (KACPR) as the initial, favoured manoeuvre [3,20,21,35,36,37]. These organisations, apart from the last two, recommend the “five by five” approach, alternating five back blows with the second manoeuvre, either five chest thrusts or five abdominal thrusts. Unique differences are only presented by the RCA, who do not specify the number of back blows’ repetitions, and the Resuscitation Council of Southern Africa (RCSA), which recommends back blows as a secondary manoeuvre [21,22].

#### 3.1.1. Effectiveness

Back blows’ effectiveness has been found to vary across different studies and may not always be adequate to relieve FBAO in paediatric patients [34].

Dunne et al.’s study of 268 paediatric patients identified that back blows were associated with an improved likelihood of FBAO resolution and survival to hospital discharge, compared to abdominal thrusts and chest thrusts (adjusted odds ratio for FBAO relief of 0.39 and 0.92, respectively, compared to back blows). Furthermore, the researchers found that back blows did not result in any intervention-related injuries, unlike abdominal thrusts and chest thrusts [38].

In Norii et al.’s study, the success rate of back blows was estimated at 10.4%, which was half as effective as other interventions, i.e., suction and abdominal thrusts. In contrast, they also observed that patients with a favourable outcome were more likely to receive back blows as the initial intervention (23.2% vs. 11.0%). This indicates a relatively low success rate if they do not immediately relieve the object as the initial manoeuvre. However, their study included only two children: one under 10 years old and one aged between 10 and 19, within a total of 300 patients. It also only included individuals who presented to the emergency department; therefore, they were more likely to have a more complicated FBAO, and did not capture those who were treated on scene and discharged without transport. Most identified participants in this study were elderly individuals with pre-existing swallowing difficulties who needed a diet of semi-solid food and thickened fluids. The findings indicated that healthcare providers may have considered suction to be a more effective option than other manoeuvres. Therefore, the findings are not representative of the overall effectiveness of back blows in treating paediatric FBAO [39].

Igarashi et al. also conducted a FBAO study that demonstrated a higher success rate compared to Norii et al.’s study (27% vs. 10.4%, respectively). Again, however, patients had a median age of 80 years and there were no children included in this study. Patients who received back blows as the first intervention had significantly better neurological outcomes after the incident than those who received other manoeuvres. No statistical significance was observed for other actions. This study’s sample size was very small. For back blows, there were only 22 identified individuals with 16 achieving a favourable neurological outcome compared to 6 with a poor neurological outcome [4].

Another study by Dunne et al. involving 124 patients who had airway clearance devices used as an FBAO manoeuvre demonstrated that back blows are widely utilised and commonly employed. This study included 55 paediatric cases. The study was not designed to assess the effectiveness of back blows; however, it is important to note that minimal adverse events were reported due to them [40].

Montana et al. described a case involving a 3-year-old girl who suffered from FBAO while consuming mozzarella cheese at school. Despite attempts by teachers to administer first aid with repeated back blows, and even the intervention of an experienced anaesthesiologist trying endotracheal intubation using a laryngoscope multiple times, they were unable to dislodge the obstruction. Postmortem examination revealed blockage at the aditus ad laringem (laryngeal inlet) [41]. It is likely that no bystander FBAO manoeuvre would have removed this obstruction.

#### 3.1.2. Safety

The harm associated with back blows is considered minimal compared to other FBAO interventions [34]. The concerns about the lack of safety in administering back blows are based on case studies, but research articles investigating the effectiveness of back blows on larger populations demonstrate no adverse effects [4,37,38,39].

Ekim-Altun arrived at similar findings in their research as Dunne et al. [38]. Their investigation revealed that 31% of mothers administered back blows to children experiencing FBAO. Other actions taken included non-recommended interventions such as forcing the child to vomit and blind finger sweeping of the mouth. These results demonstrate that even untrained parents are familiar with the technique of back blows. Although they studied a group of 163 patients, mostly infants, they did not report any adverse effects, challenging claims about the potential risks associated with back blows [42].

Dr Heimlich’s assertion that changed the world’s attitude towards back blows stated that “back blows are ineffective for choking and can drive an object deeper into the airway” [17]. This has not been substantiated since the 1980s, when back blows were applied with other manoeuvres such as mouth-to-mouth ventilation and blind finger sweeping of the mouth. These other manoeuvres are far more likely to be responsible for this complication [3]. The study proving Dr Heimlich’s thesis and comparing abdominal thrusts and back blows was conducted by Day–Crelin–DuBois in 1982 [43]. The researchers credited support from “The Dysphagia Foundation Inc.”, which later became “The Heimlich Institute”, in their acknowledgments at the end of the paper [44]. This connection between Dr Henry J. Heimlich and the Yale scientists raises concerns about a possible conflict of interest. It appears appropriate to reformulate Dr Heimlich’s thesis. The current data collectively suggest that, while any intervention carries some degree of risk, back blows remain one of the safer and more effective techniques for managing FBAO, particularly in paediatric cases [34,37,38,39,42].

The only recent study describing complications after administration of back blows, referenced by ILCOR, is the Guinane–Lee study. The research details a vascular injury in an older individual with risk factors for vascular diseases following FBAO first aid management involving chest thrusts and back blows, and they were unable to know which manoeuvre resulted in the injury. Further, the relevance of this case study to back blows, particularly in paediatrics, is limited [45].

### 3.2. Chest Thrusts

Chest thrusts squeeze the air out of the lungs by increasing intrathoracic pressure, performed by embracing the choking victim from behind and using a closed fist to press on the lower half of the sternum above the xiphoid process, as presented in Figure 3. A slight bend over the victim’s waist toward the ground should be applied.

Most guidelines recommend using chest thrusts when abdominal thrusts are not feasible for the victim due to pregnancy or the individual’s size [3,19,21]. Experts also recommend chest thrusts as a second manoeuvre to alternate with for infants. However, ANZCOR and the Australian and New Zealand divisions of SJA advocate an innovative approach and propose using chest thrusts as a secondary manoeuvre in older children, and JRC even allows the use of chest thrusts as an initial manoeuvre [20,21,46,47,48,49]. They claim it is safer with a lower potential of life-threatening complications compared to abdominal thrusts.

It is important to highlight some modifications that have been documented, or even recommended, which deviate from the standard chest thrust technique. ANZCOR guidelines recommend administering chest thrusts from the front, with the victim’s back against a firm surface like a wall or a floor [20]. Additionally, the Australian and New Zealand divisions of SJA propose supporting one hand in the middle of the victim’s back, between their scapulas, while placing the heel of the other hand on their lower sternum [46,47,48]. The reasoning behind this recommendation and which technique is superior remains unclear.

It continues to be an observable pattern, even within academic collective research, of incorrectly confusing chest thrusts with chest compressions and failing to specify the precise technique administered to the individual [4,5,38,40]. Chest compressions may have the potential to generate potentially greater force, as gravity assists the rescuer and the floor serves as a stable brace. Conversely, standard chest thrusts, performed while standing, may generate less force, as the rescuer’s own body acts as the brace and gravity does not play a significant role. Not specifying which technique form was used when reporting on FBAO interventions can potentially lead to false conclusions about the effectiveness and safety of lifesaving manoeuvres.

#### 3.2.1. Effectiveness

In 2010, ANZCOR determined that higher airway pressures could be generated by using chest thrusts rather than abdominal thrusts or back blows [20,44,45]. While the exact airway pressure values created by chest thrusts are currently unknown, standard chest compressions, used in CPR, can achieve pressure values of 40.8 ± 16.4 cmH_2_O, which may provide an adequate or similar level of pressure for chest thrusts [50]. However, chest thrusts continue to be the least researched manoeuvre, advised by the fewest first aid experts.

Shim–Park documented a case involving a 12-month-old infant who required three sets of 5 chest thrusts and 5 back blows after swallowing a foreign object. These efforts proved ineffective and the patient deteriorated into respiratory failure and cardiac arrest. The foreign body was eventually pushed into a distal bronchus with an endotracheal tube and removed later with bronchoscopy after the return of spontaneous circulation. This case, however, describes a complex and challenging scenario of a foreign body located deep in the trachea [51].

Norii et al. conducted a study which documented 59 instances of chest thrusts being administered to adult patients with FBAO. However, no cases were found where the use of chest thrusts led to successful removal of the foreign object from the airway. The research specifically focused on elderly patients and its sample size of only two children may limit the generalizability of its findings on the effectiveness of chest thrusts in treating paediatric FBAO [39].

#### 3.2.2. Safety

Chest thrusts have fewer case reports published where a traumatic injury occurs compared to abdominal thrusts. This may imply that chest thrusts could be a safer and potentially more effective intervention for the first aid management of paediatric FBAO, though the relative safety and efficacy of these techniques remains uncertain.

However, Dunne et al. reported that chest thrusts, also referred to as chest compressions in the study, showed the highest proportion of injuries. The study population encompassed both conscious and unconscious individuals, potentially diluting the results and complications of chest thrusts as an intervention [38].

It is worth mentioning the Guinane–Lee research again, which documented a single case of acute thoracic aortic dissection in an 85-year-old man with underlying conditions such as atherosclerosis, hypertension, hypercholesterolemia, and a history of significant smoking. The administration of other interventions to the patient and his vascular risk factor profile raises doubts about whether chest thrusts, back blows or choking itself is responsible [45].

### 3.3. Abdominal Thrusts

Abdominal thrusts, previously referred to as the Heimlich manoeuvre, are arguably the most frequently recommended manoeuvre in modern protocols for managing FBAO [3,19,21]. In this technique, a rescuer applies pressure to the bottom of the diaphragm from behind an individual bending forward to compress the lungs and dislodge an obstructing object from the airway by increasing intrathoracic pressure, as presented in Figure 4.

The ERC, IFRC, MFMER, SJA, and RCUK suggest using this method as a secondary technique in combination with back blows [3,23,35,36,37,52,53,54,55]. The KACPR has a similar view but agrees on administering abdominal thrusts only to children older than 8 years old. The AHA, RCSA and most RCA members advise using it as an initial or sole approach [19,21]. It is widely agreed that this method should not be used on infants due to their fragility [34]. However, an increasing number of institutions now also advise against using abdominal thrusts in children over 1 year old for similar reasons. Consequently, ANZCOR, JRC, and the Australian and New Zealand divisions of SJA have opted to substitute chest thrusts for abdominal thrusts in children [20,21,46,47,48,49]. Abdominal thrusts are perhaps the most contentious manoeuvre of the three, with the most thorough examination and numerous modifications, with the most famous called the Heimlich manoeuvre.

It is essential to recognise that abdominal thrusts may not solely involve the Heimlich manoeuvre. Despite these terms being frequently interchangeable in scientific literature, the classical Heimlich manoeuvre involves placing the fist just below the ribcage and approximately two inches above the victim’s navel with inward and upward thrusts [15,18]. However, this approach could be distressing for children. We found only one source that discusses the disparity between the two techniques [5]. There is a lack of studies demonstrating alternative techniques involving lower hand placement, flat hands, or interlocked fingers. Furthermore, several alternative methods have been reported to deviate or evolve from the standard abdominal thrust technique, including self-administering the manoeuvre manually and with the use of a chair, or performing the procedure while facing the victim [54].

#### 3.3.1. Effectiveness

Langhelle et al. demonstrated in their study with adult cadavers that abdominal thrusts can produce an average peak airway pressure of approximately 26.4 ± 19.8 cm H_2_O [50].

Norii et al. conducted a study involving a large group of patients with in-hospital FBAO. Abdominal thrusts were administered to 24 individuals, successfully removing the foreign body in 5 cases, resulting in a 20.8% success rate. It is crucial to emphasise that this was an assessment of abdominal thrusts as the initial response and primarily involved older patients with pre-existing swallowing issues who required semi-solid foods and thickened fluids [39].

#### 3.3.2. Safety

Researchers found that the average peak stomach pressure during abdominal thrusts can reach 57 ± 17 cmH_2_O, with potentially higher values in children [13]. This explains the effectiveness of the Heimlich manoeuvre and its potential to cause direct injury to abdominal organs or tissues, with gastric perforation being the primary concern [55,56].

Ebrahimi and Mirhaghi’s review suggests that an adult can exert excessive force with an abdominal thrust, which may cause fatal internal injury in children. They documented 48 cases of severe complications following the Heimlich manoeuvre, a notable portion involved patients under 18 years old. Surgery successful treated the injury in 25% of cases with organ damage, while the remaining cases resulted in fatalities. The authors assert that life-threatening injuries related to the Heimlich manoeuvre indicate a need for a safer alternative procedure [57].

Koss et al. presented a case of cervical oesophageal perforation in a 16-year-old boy resulting from the Heimlich manoeuvre. The patient experienced symptoms and was eventually treated via a successful but difficult surgical repair. However, this injury is linked to a high mortality rate and requires coordinated, prompt care from multiple disciplines [58].

Wang et al. reported a story of a 52-year-old man who suffered from choking and loss of consciousness. After receiving abdominal thrusts from an inexperienced first aid provider, the obstruction was cleared, but he later experienced upper abdominal discomfort. An examination revealed gastrointestinal haemorrhage and severe damage to the anterior wall of the right ventricle, which proved fatal. The authors reviewed the literature on abdominal thrust complications and found 11 cases of gastric ruptures, 10 cases of aortic injuries, and 2 cases of pancreatic injuries, noting that these injuries also occur in younger individuals [55].

Basile et al. documented a situation where oesophageal rupture occurred resulting from the Heimlich manoeuvre. They indicated that abdominal thrusts could lead to traumatic injury of the gastrointestinal tract, pneumomediastinum, rib fracture, diaphragm rupture, acute thrombosis of an abdominal aortic aneurysm, and mesenteric laceration. Abdominal injuries were the most frequently observed complications, particularly oesophageal and gastric wall rupture. Moreover, they emphasised that abdominal thrusts are difficult to apply correctly and untrained individuals performing this manoeuvre may increase the likelihood of serious complications [59].

Pawlukiewicz et al. discussed a case involving a 56-year-old female patient admitted to the hospital with abrupt pain in her right foot following an episode of FBAO that was resolved using the Heimlich manoeuvre. An examination revealed distal arterial occlusion caused by cholesterol embolization syndrome, highlighting a previously unrecognised complication of this manoeuvre that can lead to substantial morbidity [60].

Herman et al. documented a case of an 85-year-old woman who experienced choking due to food ingestion. Medical personnel performed the Heimlich manoeuvre, successfully removing the obstruction but causing complications, including stomach herniation into the right lower chest. The patient required emergency surgery for hernia repair and subsequently faced challenges during her hospital stay, including peritonitis, fascial dehiscence, and necrotizing fasciitis, before being discharged home [61].

A comparable scenario was described by Truong et al. involving a patient with a substantial incarcerated hiatal hernia, necessitating surgical intervention for reduction and gastropexy. Despite initial improvement, the patient subsequently developed septic shock and severe malnutrition, resulting in an extended hospitalisation involving multiple surgeries and intubations. Similar cases exhibited unfavourable results, including mortality or additional surgeries to address various complications. Additionally, they referenced a case series detailing adverse outcomes associated with Heimlich manoeuvre complications: 13 out of 41 patients affected experienced fatalities, while another 17 required surgeries to repair different organs [62].

Lee et al. presented a case of a 67-year-old man without widespread medical conditions, who arrived at the emergency department with paralysis on his left side shortly after an emergency medical technician performed the Heimlich manoeuvre to clear a blocked airway caused by a piece of meat. The chest CT scan showed that he had a Stanford type A aortic dissection and an obstruction in the right innominate artery. An urgent surgical procedure was performed to repair the aorta using grafting, and he was discharged from the hospital without complications [63].

Tashtoush et al. reported a case involving an 84-year-old man who was taken to the emergency department after choking at a restaurant, followed by unsuccessful Heimlich manoeuvre attempts. Although a large piece of steak causing airway obstruction was successfully removed, the patient remained hypotensive and needed ongoing hemodynamic support. Subsequent laboratory tests conducted within 24 h of aspiration revealed a significant decrease in haemoglobin levels. A computed tomography scan of the abdomen and pelvis indicated a lacerated liver with a substantial subcapsular haematoma draining into the pelvis [64].

Bouayed et al. reported a case of a 45-year-old mentally disabled woman who experienced acute respiratory distress from choking on a large piece of chicken. After the Heimlich manoeuvre, she developed subcutaneous emphysema. CT scans showed a 3 cm bone fragment in the oesophagus and widespread emphysema. Endoscopy removed the bone, revealing a 3 mm tear. Despite initial treatment with antibiotics and a nasogastric tube, she developed fever and respiratory distress, leading to the emergency drainage of an abscess. Subsequent care included antibiotics and drainage, resulting in full recovery. The causation, however, remains unclear, as the incident was likely attributable to choking rather than the Heimlich manoeuvre, or at the very least, the specific cause remains uncertain [65].

### 3.4. Other Techniques

Although this review focuses on rescue manoeuvres used to relieve obstruction, it is important to acknowledge some alternative techniques that may be utilised in FBAO first aid cases or are not widely recognised by resuscitation councils.

Cough encouragement remains widely recommended in various guidelines and continues to be an important technique in first aid for conscious, choking children with an effective cough, which suggests a partial obstruction [3,19,20,21,22,23]. Despite its long-standing recognition, it has not been subjected to extensive recent research, which limits the availability of current data on its effectiveness and safety in paediatric patients. The belief is that coughing can manage the choking individual most effectively and safely because it does not require the application of an external force which risk secondary injuries as other FBAO interventions. However, there are several research gaps associated with its efficacy. In children, the lower respiratory strength due to their smaller thoracic musculature may be insufficient to generate the airway pressures needed to effectively clear the obstruction. Relying on cough encouragement in critical situations could delay more effective and relatively safe rescue manoeuvres, such as back blows, which could also be combined with the choking individual’s own coughing efforts. Furthermore, it may be challenging for a layperson to accurately assess the efficacy of the cough. Encouraging coughing may also create a false sense of security, further delaying necessary medical intervention. Additionally, young children may not comprehend instructions or may become uncooperative due to the distressing nature of the situation, making cough encouragement impractical. There is also a lack of evidence regarding the proper body position for the choking individual during coughing. Positioning the individual in a forward-leaning or knee-chest position may reduce the risk of inadvertently pushing the foreign object deeper, which could block the airway completely, and may improve the effectiveness of the procedure by utilising gravity to assist in clearing the obstruction. These considerations warrant further scientific investigation.

Blind finger sweep of the oropharynx is generally advised against by organisations such as the ERC, with some guidelines omitting the technique entirely from their recommendations [3,19,20,21,22,23]. It is rarely the focus of contemporary reports and lacks recent updates regarding its safety and effectiveness. Concerns include potentially fatal pharyngeal trauma, traumatic epiglottitis, or pushing the object further into the airway as well as its ineffectiveness in cases of obstruction localised in the trachea or larynx, as noted in studies conducted a decade or more ago [66,67,68,69]. A 2016 case study by Mori and Inoue describes a 1-year-old boy who began choking and coughing after swallowing a coin. The mother tried to remove it by performing a blind finger sweep but was unable to retrieve it. The child was taken to the hospital, where he showed mild gagging but no respiratory issues. X-rays revealed a nasopharyngeal foreign body, which was successfully removed under sedation by an otolaryngologist. The child was discharged without complications [70]. Similarly, Vunda and Vandertuin reported a case in which a 9-month-old girl developed respiratory distress after playing with a cufflink. Her mother attempted a blind finger sweep but failed to retrieve the object. At the emergency department, the child showed no respiratory distress, and an X-ray did not reveal the cufflink. However, additional imaging identified the cufflink lodged in the nasopharynx, and it was subsequently removed under general anaesthesia [71]. These two cases demonstrate instances where a rushed and unrecommended technique may have inadvertently pushed the foreign body from the oropharynx to the nasopharynx. It is important to emphasise that this may have led to a temporary restoration of respiratory function, but the outcome remained suboptimal. Given this weak scientific evidence, blind finger sweep might still have a potential to be appropriate in specific scenarios (e.g., as a method of last resort); further research could clarify its role in first aid for FBAO, particularly in paediatric cases.

Airway clearance devices, also referred to as anti-choking suction devices, provide a clear example of how the topic of FBAO should be actively investigated. They act as non-powered, negative pressure device that attempts to relieve a FBAO from above, instead of creating a force distal to the obstruction as in traditional FBAO manoeuvres. Due to their novelty, these devices have become a frequent subject of research, with studies focusing on their effectiveness, safety, and potential application [40,72,73,74,75,76,77,78,79,80]. The initial systematic review on the topic in 2020 found the available data on these devices were limited, and provided insufficient evidence to either support or discourage their use [76]. Since then, a number of subsequent studies have been published. Dunne et al. conducted two studies assessing the effectiveness and safety of the two widely recognised devices, LifeVac^©^ (LifeVac LLC, Nesconset, New York, NY, USA) and Dechoker^©^ (Dechoker LLC, Wheat Ridge, CO, USA). Data were collected from 371 patients, with a significant 58% of the studied population being children. In 361 cases, the airway clearance device was the final intervention before the successful resolution of FBAO symptoms. Its use was associated with only a few, generally mild, adverse events [40,72]. While the findings were promising, they should be interpreted cautiously due to study limitations, including self-reporting biases within the sample population and reliance on non-medical personnel for FBAO diagnosis. The study by Bhanderi and Hill found consistent results, with the Dechoker^©^ reported to have successfully removed the obstruction in 26 out of 27 adult cases with few complications or adverse events reported [73]. Additionally, research by Cardalda-Serantes et al. and Carballo-Fazanes et al. indicates that untrained health science students and paediatric residents were able to effectively learn and utilise airway clearance devices more effectively than the current FBAO algorithms [74,75]. To date, research largely relies on self-reported data and studies involving non-medical personnel. While this may raise concerns about the reliability and generalizability of these findings, these data are valuable given that FBAO incidents often occur in situations where only first aid measures are applied without professional intervention. An increasing number of sources suggest that airway clearance devices may have potential and could be considered in choking emergencies when standard protocols prove inadequate.

### 3.5. Take-Home Message

This review represents the most comprehensive evaluation of recent literature conducted to date. Table 3 was created to enhance the visualisation of reviewed scientific articles. It should be noted that the oldest article examined dates back to 2015, indicating that the reported data are current and up to date.

As shown in the table above, despite rescue manoeuvres for severe FBAO in paediatric patients being widely recommended in various guidelines for decades as the best and only first aid treatment, there is a notable lack of valuable data and research on their effectiveness and safety. Only nine studies have reported paediatric cases, often involving small, non-representative samples.

## 4. Conclusions

Knowledge gathered from relevant research, analysis, and case studies on paediatric FBAO is scarce and insufficient, which explains why significantly differing guidelines are being created worldwide. Supplementing the data with adult FBAO cases, which are also limited, may lead to a subjective belief that back blows should be used as an initial manoeuvre due to their safety, followed by chest thrusts for generating high airway pressure with low reported complications. Abdominal thrusts were found to have the highest number of studies reporting the potential for trauma and may result in lower average peak airway pressure. However, this is likely due to the long-standing belief in the effectiveness of these rescue manoeuvres, which is often taken for granted, along with the recent focus on safety in research. This trend is especially apparent in abdominal thrusts. Future recommendations should focus on evaluating the optimal balance between safety and effectiveness.

## 5. Future Directions

We conclude that providing specific recommendations would be futile or even harmful without acquiring the necessary, currently non-existent, scientific evidence. FBAO is a common and significant concern, with over 75% of choking occurrence rate in children younger than 3 years [2,81]. Epidemiological data on children under 16 years old reveal thousands of documented fatalities every year, positioning FBAO as a leading cause of unintentional death [2]. It is widely recognised as one of the primary causes of death among paediatric populations and leads to an estimated annual 300 to 600 fatal incidents in developed countries each, with double the proportion occurring in undeveloped countries [81,82,83]. Despite increased awareness, the frequency of FBAO among children has consistently risen over an extended period [84].

It is shocking that so many children die each year due to choking, yet there is a worldwide shortage of data reports and research focusing on emergency interventions in FBAO, particularly in this specific demographic group, which demands special attention. This state of affairs is unacceptable. We propose a solution for data collection and flow, which is an original project for collecting high-quality data that can inform future recommendations and depicted in Figure 5.

The FBAO Database functions by reporting cases documented by the healthcare provider, responsible for patients who experienced choking. Cases can be reported to the global or even local FBAO Database, whether in an ambulance or emergency department. Any patient with severe airway obstruction or who has received external force manoeuvres such as back blows, chest thrusts, and abdominal thrusts should be included. If the patient undergoes surgery as part of their treatment, or in a worst-case scenario, if choking results in fatality, it will be the operating doctor’s or any care team member’s responsibility to document and report the case.

The document responsible for reporting these cases to the FBAO Database is called a Chokelist. Our suggested Chokelist is presented in Figure 6 and comprises sections that are intended to present data and support EBM, based on the case studies examined in the review. The first part outlines the patient’s details, such as age, sex, and medical history. The next section discusses the specific incident, symptoms, extent of obstruction, type of foreign object causing the blockage (e.g., food or toy), and first aid provided by bystanders. The third part focuses on the patient’s medical care and their outcome, encompassing invasive rescue manoeuvres to alleviate ongoing obstruction and alternative treatments like surgical interventions and anaesthesia procedures. Finally, it concludes with an account of subsequent developments in the patient’s condition and their final outcome.

Our hope is that a standardised reporting tool will significantly increase available data and improve the understanding of paediatric FBAO, leading to the development of high-quality, evidence-based, dependable guidelines comprising effective and safe rescue manoeuvres.

### Limitations

This study has limitations, including a small sample size due to the limited availability of material in the literature. The data were supplemented by adult research, which may not accurately reflect the specific challenges and nuances of paediatric foreign body airway obstruction. The collected data vary in quality and diverse characteristics, making it potentially non-generalizable to all healthcare settings. Most of the data comes from case studies and literature reviews, which also have limitations such as prejudice toward the discussed issue and a lack of generalizability. Additionally, being a narrative review, this study is subject to some disadvantages, like potential bias from the authors of retrieved studies and article reviewers. In order to address the existing gaps in the data, which contribute to discordant recommendations, future research should periodically incorporate systematic reviews and meta-analyses, while also increasing the number of data-gathering studies. This effort would benefit from the structured use of our Chokelist and data collection flow concept to enhance consistency and comprehensiveness in reporting.

## Figures and Tables

**Figure 1 medicina-60-01827-f001:**
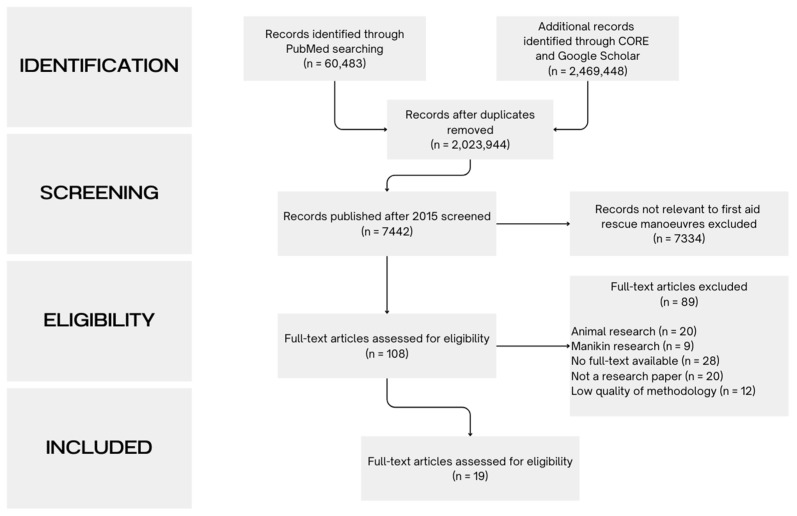
Preferred Reporting Items for Systematic Reviews and Meta-Analyses (PRISMA) flow diagram.

**Figure 2 medicina-60-01827-f002:**
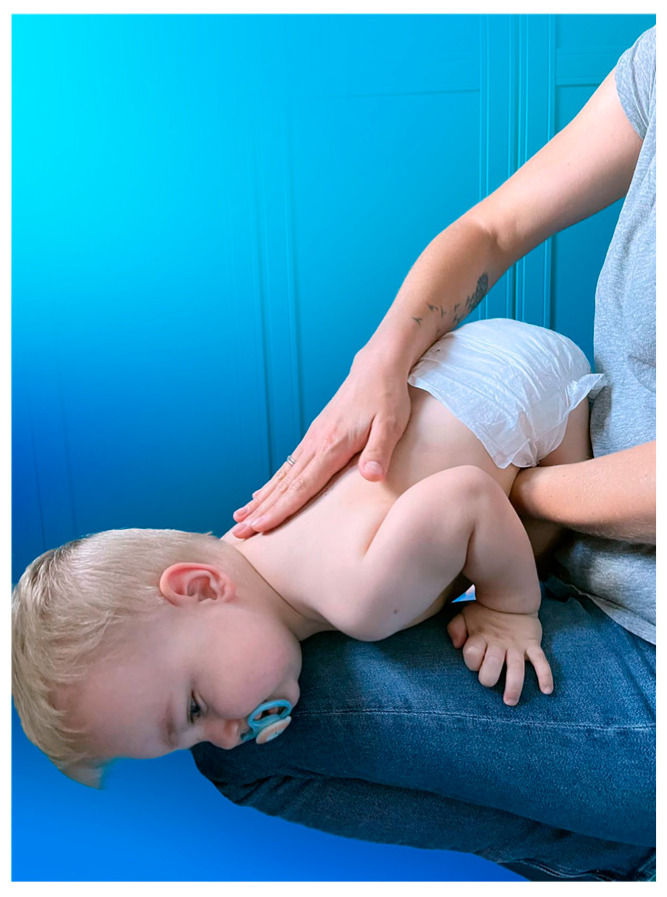
The standard back blows method presented on a 16-month-old boy by his mother (the pacifier was used solely to soothe the child for the accurate demonstration of the rescue manoeuvre).

**Figure 3 medicina-60-01827-f003:**
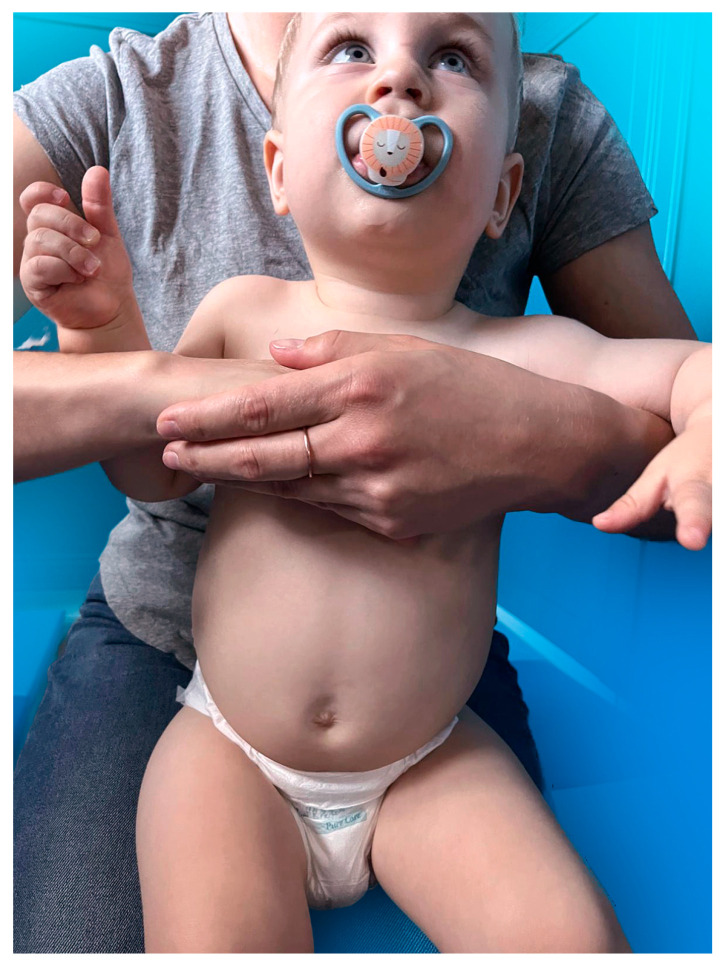
The standard chest thrusts method presented on a 16-month-old boy by his mother (the pacifier was used solely to soothe the child for the accurate demonstration of the rescue manoeuvre).

**Figure 4 medicina-60-01827-f004:**
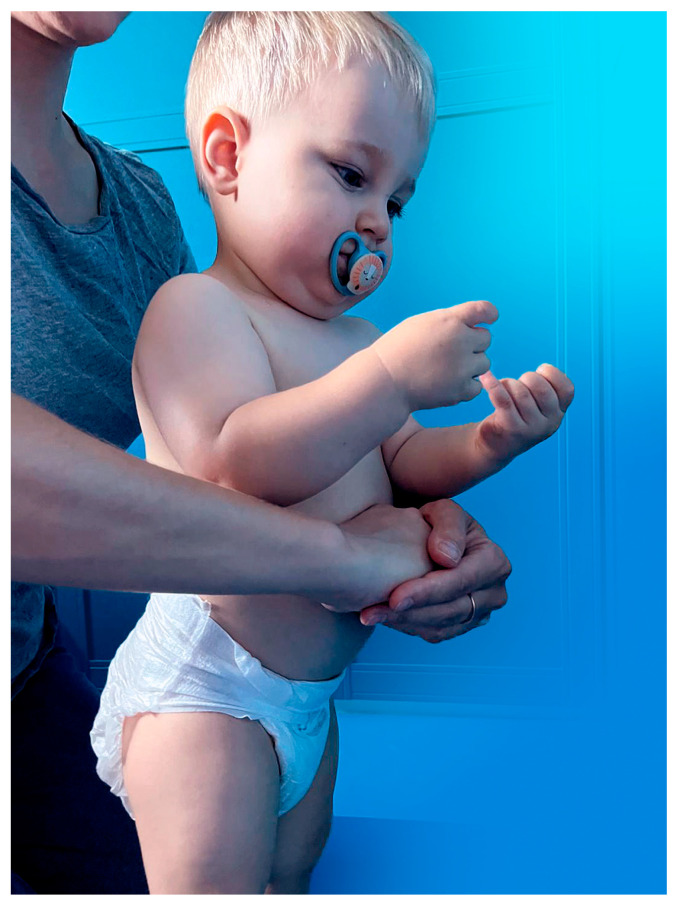
The standard abdominal thrusts method presented on a 16-month-old boy by his mother (the pacifier was used solely to soothe the child for accurate demonstration of the rescue manoeuvre).

**Figure 5 medicina-60-01827-f005:**
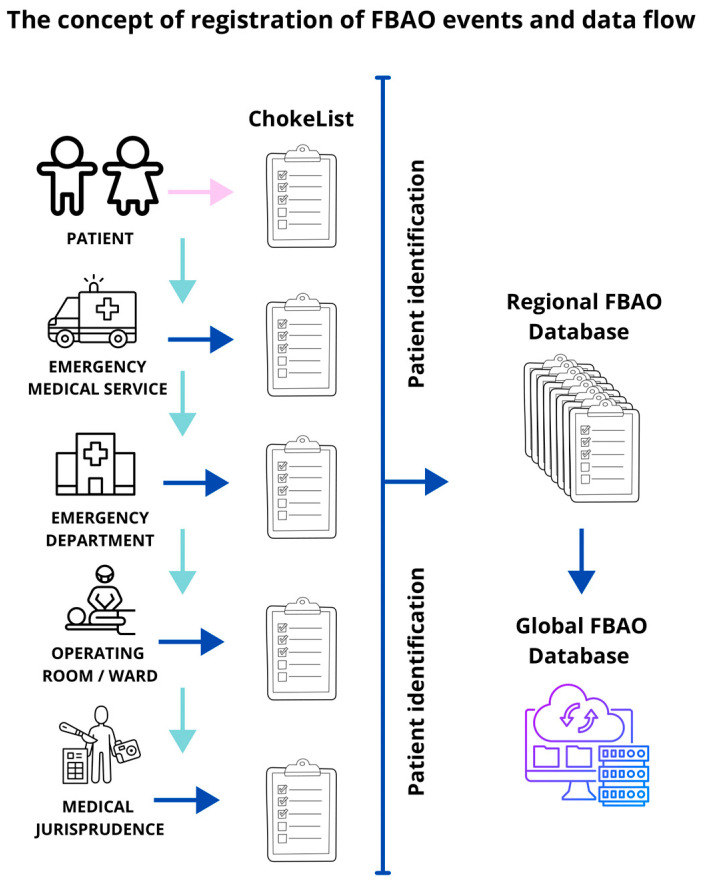
The original concept for data collection and flow for foreign body airway obstruction cases in children.

**Figure 6 medicina-60-01827-f006:**
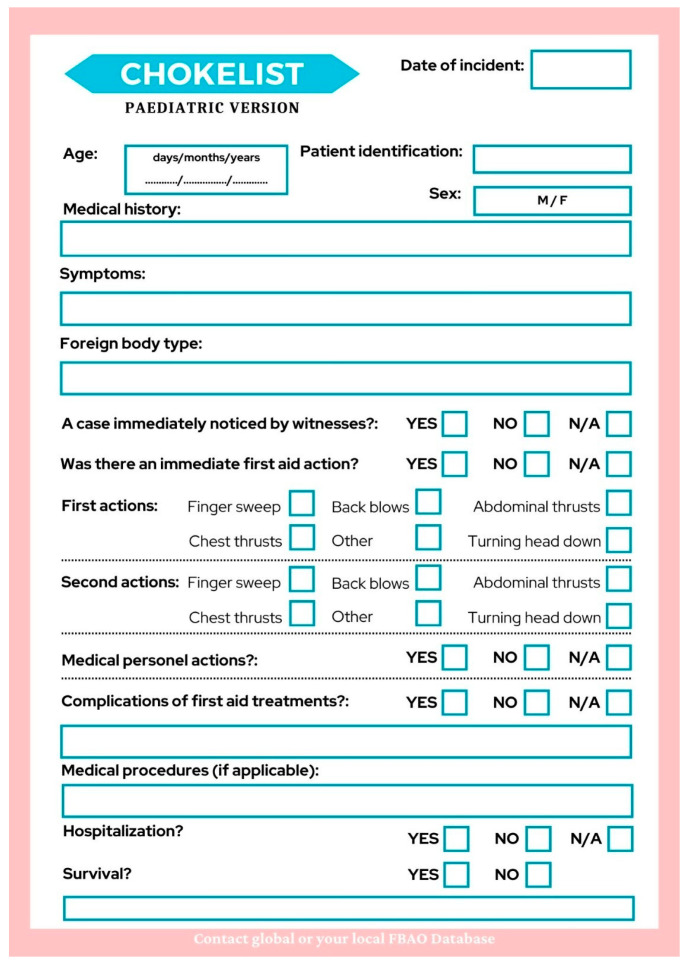
Paediatric Chokelist—original document responsible for reporting foreign body airway obstruction cases.

**Table 1 medicina-60-01827-t001:** A comparison of the differences in recommended manoeuvres and their sequence for managing foreign body airway obstruction in children older than 1 year old, who are conscious with an ineffective cough [3,19,20,21,22,23].

Guidelines	Back Blows	Chest Thrusts	Abdominal Thrusts
ERC ^1^	first	not recommended	second
AHA ^2^	not recommended	not recommended	only
IFRC ^3^	first	not recommended	second
MFMER ^4^	first	not recommended	second
ANZCOR ^5^	first	second	not recommended
JRC ^6^	first or second (order does not matter)	first or second (order does not matter)	not recommended
KACPR ^7^	only (if child < 8 yrs. old) first (if child ≥ 8 yrs. old)	not recommended	second (if child ≥ 8 yrs. old)
RCA ^8^	not recommended	not recommended	only
RCSA ^9^	second	not recommended	first
SJA ^10^	first	second (Australia and New Zealand)	second
RCUK ^11^	first	not recommended	second

^1^ ERC—European Resuscitation Council, ^2^ AHA—American Heart Association, ^3^ IFRC—International Federation of Red Cross and Red Crescent Societies, ^4^ MFMER—Mayo Foundation for Medical Education and Research, ^5^ ANZCOR—Australian and New Zealand Committee on Resuscitation, ^6^ JRC—Japanese Resuscitation Council, ^7^ KACPR—Korean Association of Cardiopulmonary Resuscitation, ^8^ RCA—Resuscitation Council of Asia, ^9^ RCSA—Resuscitation Council of Southern Africa, ^10^ SJA—Saint John Ambulance, ^11^ RCUK—Resuscitation Council UK.

**Table 2 medicina-60-01827-t002:** A comparison of the algorithm differences in manoeuvres recommended by Canadian training agencies and their sequence for managing foreign body airway obstruction in children older than 1 year old, who are conscious with an ineffective cough [24,28].

Training Agencies	Back Blows	Chest Thrusts	Abdominal Thrusts
CRC ^1^	Any combination of back blows, chest thrusts or abdominal thrusts		
CSP ^2^	Any combination of back blows, chest thrusts or abdominal thrusts		
HSF ^3^	not recommended	not recommended	only
RLSS ^4^	Any combination of back blows, chest thrusts or abdominal thrusts		
SJA ^5^	first	not recommended	second

^1^ CRC—Canadian Red Cross, ^2^ CSP—Canadian Ski Patrol, ^3^ HSF—Canadian Heart and Stroke Foundation, ^4^ RLSS—Royal Life Saving Society Canada, ^5^ SJA—St. John Ambulance Canada.

**Table 3 medicina-60-01827-t003:** Recent studies investigating the safety and efficacy of back blows, chest thrusts, and abdominal thrusts.

Author	Year	Type of Research	Reported Paediatric Case(s)	Rescue Manoeuvre(s) Studied
Shim-Park [51]	2024	case study	X	back blows and chest thrusts
Dunne et al. [38]	2024	observational cohort study	X	all three
Norii et al. [39]	2023	systematic review	X	all three
Ekim-Altun [42]	2023	retrospective review	X	back blows
Basile et al. [59]	2023	case study and literature review		abdominal thrusts
Igarashi et al. [4]	2022	multicentre retrospective observational study		all three
Dunne et al. [40]	2022	retrospective study	X	all three
Wang et al. [55]	2022	case study and literature review		abdominal thrusts
Pawlukiewicz et al. [60]	2021	case study		abdominal thrusts
Couper et al. [34]	2020	systematic review	X	all three
Montana et al. [41]	2020	case study and literature review	X	back blows and abdominal thrusts
Gutierrez-Strickland [56]	2020	case study		abdominal thrusts
Ebrahimi et al. [57]	2019	systematic review	X	abdominal thrusts
Lee et al. [63]	2019	case study		abdominal thrusts
Guinane-Lee [45]	2018	case study		back blows and chest thrusts
Koss et al. [58]	2018	case study	X	abdominal thrusts
Herman et al. [61]	2018	case study		abdominal thrusts
Truong et al. [62]	2017	case study		abdominal thrusts
Bouayed [65]	2015	case study		abdominal thrusts

## Data Availability

No new data were created or analysed in this study. Data sharing is not applicable to this article.

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
