# Peer review of "Do We Actually Help Choking Children? The Quality of Evidence on the Effectiveness and Safety of First Aid Rescue Manoeuvres: A Narrative Review"

_medicina, 2024, doi:10.3390/medicina60111827_

Round 1
Reviewer 1 Report
Comments and Suggestions for Authors
Few of my concers are as follows
1. As the inconsistencies in FBAO management guidelines across various global resuscitation organizations, what do you believe are the main barriers preventing the development of a universally accepted standard for managing pediatric FBAO?
2. What are the future research that could address the current gaps in data that contribute to the discordant recommendations, particularly for children older than 1 year old with an ineffective cough?
3. The child’s mouth is closed by a pacifier, how would the effectiveness of back blows in dislodging a foreign body be impacted, and should there be specific guidelines on handling such situations during FBAO management in toddlers? It is there in all pictures.
4. Due to potential risks associated with back blows, including the assertion by Dr. Heimlich, how do recent studies and your findings challenge or support these claims? Are there any updated safety recommendations based on current research, especially in pediatric cases?
5. Your review highlights a variety of interventions and alternative techniques for managing FBAO, including the novel use of airway clearance devices. However, the reliance on self-reported data and studies involving non-medical personnel raises concerns about the reliability and generalizability of these findings. Justify.
6. while you discuss the limitations of cough encouragement and blind finger sweeps, it would be valuable to explore whether these methods have specific scenarios where they might still be appropriate. Could further research clarify their role in first aid for FBAO, particularly in cases where other techniques are unavailable or impractical?
7. Preferred Reporting Items for Systematic Reviews and Meta-Analyses to give
Author Response
Response to Reviewer 1 Comments
Comments 1: Few of my concers are as follows
Response 1: Thank you for taking your valuable time to review our manuscript and provide the opportunity for us to address your concerns. The feedback enhanced the overall quality of the article. Please find the detailed responses below and the corresponding revisions/corrections in track changes in the re-submitted files.
Comments 2: As the inconsistencies in FBAO management guidelines across various global resuscitation organizations, what do you believe are the main barriers preventing the development of a universally accepted standard for managing pediatric FBAO?
Response 2: Thank you for your insightful question regarding the barriers to developing a universally accepted standard for managing pediatric FBAO. As stated in paragraphs 607-609, "Knowledge gathered from relevant research, analysis and case studies on paediatric FBAO is scarce and insufficient, which explains why significantly differing guidelines are being created worldwide."
Comments 3: What are the future research that could address the current gaps in data that contribute to the discordant recommendations, particularly for children older than 1 year old with an ineffective cough?
Response 3: Thank you for your question regarding future research to address current data gaps contributing to discordant recommendations, especially for children over 1 year old with an ineffective cough. In response, we have added suggested future research directions to the limitations section, specifically in paragraphs 676 to 681. These additions aim to highlight potential studies that could clarify guidelines and improve consistency in recommendations. We appreciate your feedback, which helped us enhance the discussion on research gaps.
Comments 4: The child’s mouth is closed by a pacifier, how would the effectiveness of back blows in dislodging a foreign body be impacted, and should there be specific guidelines on handling such situations during FBAO management in toddlers? It is there in all pictures.
Response 4: Thank you for your thoughtful question regarding the use of a pacifier in the images. We want to emphasize that the pacifier is not meant to represent a foreign body or a realistic clinical scenario, so we added a proper explanation under each of these figures (in paragraphs 227, 323 and 396). The pacifier was used to help soothe the child during the photoshoot, which is important as it allows for a proper representation of the rescue maneuvers’ techniques. Given that these are particularly valuable figures since the toddler is one of the youngest children on whom rescue maneuvers have been depicted in the gathered literature, we believe this approach adds valuable context to the discussion of first aid in choking.
Comments 5: Due to potential risks associated with back blows, including the assertion by Dr. Heimlich, how do recent studies and your findings challenge or support these claims? Are there any updated safety recommendations based on current research, especially in pediatric cases?
Response 5: Thank you for your attentive question regarding the potential risks associated with back blows, including Dr. Heimlich's assertions. We have expanded paragraphs 308-310 to include a detailed justification based on recent studies, which suggest that the harm associated with back blows is minimal compared to other FBAO interventions, particularly in pediatric cases. This addition provides further context and supports updated safety recommendations in line with current research findings. We appreciate your guidance, which has helped us strengthen this section of our manuscript.
Comments 6: Your review highlights a variety of interventions and alternative techniques for managing FBAO, including the novel use of airway clearance devices. However, the reliance on self-reported data and studies involving non-medical personnel raises concerns about the reliability and generalizability of these findings. Justify.
Response 6: Thank you for your reflective comment regarding the reliance on self-reported data and studies involving non-medical personnel, and how these factors may affect the reliability and generalizability of findings. In response, we have expanded our discussion of these limitations in paragraphs 586-590 to provide a more detailed explanation and a clear justification of these issues and their potential impact on the interpretation of results. We appreciate your suggestion, which has helped us to further clarify this critical aspect of the review.
Comments 7: while you discuss the limitations of cough encouragement and blind finger sweeps, it would be valuable to explore whether these methods have specific scenarios where they might still be appropriate. Could further research clarify their role in first aid for FBAO, particularly in cases where other techniques are unavailable or impractical?
Response 7: Thank you for your valuable feedback on the discussion of cough encouragement. We clarify in paragraphs 515-517 that cough encouragement is widely recommended in various guidelines and continues to be an essential technique in first aid for conscious, choking children who exhibit an effective cough, indicating a partial obstruction. This approach highlights the technique’s appropriate application in specific scenarios where it can be safely employed. Furthermore, we would like to thank you for your insightful comment regarding the potential scenarios in which blind finger sweeps might still be appropriate in FBAO management. In response, we have expanded the discussion on this topic in paragraphs 560-563, exploring specific contexts where blind finger sweeps could be considered. We believe this addition helps clarify the nuanced role these methods could play in certain first aid situations, and we appreciate your suggestion to address this aspect.
Comments 8: Preferred Reporting Items for Systematic Reviews and Meta-Analyses to give
Response 8: Thank you for your suggestion regarding the placement of the Preferred Reporting Items for Systematic Reviews and Meta-Analyses. We have created a PRISMA flow chart, located in paragraph 203, to clearly define and present how articles were selected, included, and excluded from the review. In response to your feedback, we will relocate the flow chart to the "Materials and Methods" chapter to enhance clarity and ensure it aligns with the organization of the review. We appreciate your guidance on this structural improvement and hope that these changes will enhance the clarity and the quality of our manuscript.
Reviewer 2 Report
Comments and Suggestions for Authors
dear Authors, thank you for your work. I read it and it's fashionable. I think that the title reassume very well the topic of the article and the topic original or relevant to the field. we have few evidence-based articles in this field this manuscript address a specific gap in the field.
Compared with other published material, the paper analyzed the literature, and the authors should add the Prospero number, the inclusion/exclusion criteria and the flow chart in the "methodology" section, part of the results are in the background section.
The references appropriate.
Author Response
Response to Reviewer 2 Comments
Comments 1: dear Authors, thank you for your work. I read it and it's fashionable. I think that the title reassume very well the topic of the article and the topic original or relevant to the field. we have few evidence-based articles in this field this manuscript address a specific gap in the field.
Response 1: Thank you very much for your time and valuable comments. We are convinced that the changes introduced thanks to them had a very positive impact on the quality and clarity of the article. Please find the detailed responses below and the corresponding revisions/corrections in track changes in the re-submitted files.
Comments 2: Compared with other published material, the paper analyzed the literature, and the authors should add the Prospero number, the inclusion/exclusion criteria and the flow chart in the "methodology" section.
Response 2: Thank you for your suggestion regarding the inclusion of the PROSPERO registration number. We would very much like to include this information in our manuscript; however, the Prospero database overview information indicates that our review doesn’t qualify for this registration. We appreciate your understanding and support in this matter.
Thank you for your comment regarding the inclusion and exclusion criteria. We would like to clarify that the criteria for including articles in our review are detailed in paragraphs 193-196, which state: “Articles were included based on specific criteria: relevance to first aid procedures for paediatric FBAO, methodological quality of the research, and recency of publication with a focus on articles published from 2015.”. We appreciate your attention to this aspect, as it underscores the rigorous approach taken in our review.
Thank you for your suggestion regarding the placement of the flow chart. We have created a PRISMA flow chart, located in paragraph 203, to clearly define and present how articles were selected, included, and excluded from the review. In response to your feedback, we will relocate the flow chart to the "Materials and Methods" chapter to enhance clarity and ensure it aligns with the organization of the review. We appreciate your guidance on this structural improvement.
Comments 3: part of the results are in the background section.
Response 3: Thank you for your feedback regarding the placement of certain information in the background section. We acknowledged your point regarding paragraphs 195-199, which discuss the search for guidelines in the methodology. To enhance clarity, we will delete the word "additionally" and move these lines to the beginning of the "Materials and Methods" section. This change will emphasize that the search for recommendations was the first part of our screening process, aimed at identifying and demonstrating the existing research gap. We appreciate your valuable suggestions, which will help improve the overall clarity and structure of the manuscript.
Comments 4: The references appropriate.
Response 4: Thank you for reviewing our references.
Reviewer 3 Report
Comments and Suggestions for Authors
Thank you for offering the chance to review the manuscript entitled" Do we actually help Choking children? The quality of Evidence on the Effectiveness and safety of first aid rescue maneuvers: a narrative review". I want to congratulate the authors for this very interesting and well-written manuscript. The subject is very interesting, and the review is thoroughly documented on literature search.
My only recommendation would be to organize a little bit different the sections of the article:
- 3. Results could become Results and Discussions and there is a long discussion about the published literature studies
- 3.6. Future directions - including the part "Our solution - Chokelist with data collection and Flow". This specific part is a future plan/direction that has recommendations of being implemented in many other medical facilities. It does not necessarily represent a conclusion of the article.
4. Conclusion - 1-2 ideas that conclude about the aims of the study formulated in the Introduction section. "We aim to conduct a narrative review of recent literature assessing the effectiveness and safety of standard first aid maneuvers used for saving children for choking and identify the gaps in evidence and knowledge in order to outline the direction of future research.", maybe the importance of guardian supervision, avoidance of small objects, etc.
The last paragraph is better fitted to Future directions rather than Conclusions.
Author Response
Response to Reviewer 3 Comments
Comments 1: Thank you for offering the chance to review the manuscript entitled" Do we actually help Choking children? The quality of Evidence on the Effectiveness and safety of first aid rescue maneuvers: a narrative review". I want to congratulate the authors for this very interesting and well-written manuscript. The subject is very interesting, and the review is thoroughly documented on literature search. My only recommendation would be to organize a little bit different the sections of the article:
Response 1: Thank you very much for your time and valuable comments. Thanks to the reviewers’ great comments, we have revised the article and edited it for greater clarity. The comments enhanced the overall quality of the article. Please find the detailed responses below and the corresponding revisions/corrections in track changes in the re-submitted files.
Comments 2: Section 3. Results could become Results and Discussions and there is a long discussion about the published literature studies
Response 2: Thank you for your observation. We agree that the results section includes extensive discussion of relevant literature. In response to your suggestion, we have retitled this section to “Results and Discussion”, in paragraph 218, to better reflect its content.
Comments 3: Section 3.6. Future directions - including the part "Our solution - Chokelist with data collection and Flow". This specific part is a future plan/direction that has recommendations of being implemented in many other medical facilities. It does not necessarily represent a conclusion of the article.
Response 3: Thank you for your valuable feedback regarding the placement of the Future Directions section. We have added “Future Directions”, in paragraph 619, after the Conclusions section. This placement enhances the article's structure and provides a clear transition from the conclusions to future plans. Additionally, this change addresses comments #5 that "The last paragraph is better fitted to Future Directions rather than Conclusions." We trust this approach will meet your expectations.
Comments 4: Section 4. Conclusion - 1-2 ideas that conclude about the aims of the study formulated in the Introduction section. "We aim to conduct a narrative review of recent literature assessing the effectiveness and safety of standard first aid maneuvers used for saving children for choking and identify the gaps in evidence and knowledge in order to outline the direction of future research.", maybe the importance of guardian supervision, avoidance of small objects, etc.
Response 4: Thank you for this feedback and suggestion to incorporate conclusions around the aims of first aid maneuvers specifically related to choking in children, such as the importance of guardian supervision and object avoidance. While these topics are indeed critical to pediatric safety, they extend beyond the primary objectives of our review.
Our study's focus, as established in the Introduction, was a critical assessment of the available literature on the safety and effectiveness of existing first aid maneuvers for pediatric foreign body airway obstruction (FBAO). Given the lack of high-quality data, our conclusions emphasized the need for further research and improved evidence-based guidelines for current maneuvers, specifically aimed at addressing the gaps in literature around their safety and effectiveness. We believe this focus aligns directly with our stated aim to review evidence surrounding first aid maneuvers rather than broader prevention topics. We appreciate the importance of these areas but consider them outside the scope of this review. Please let us know if further clarification is needed.
Comments 5: The last paragraph is better fitted to Future directions rather than Conclusions.
Response 5: Thank you for this great suggestion, please see our earlier response under Response 3.
Round 2
Reviewer 1 Report
Comments and Suggestions for Authors
Well revised